# T-RAIM Approaches: Testing with Galileo Measurements

**DOI:** 10.3390/s23042283

**Published:** 2023-02-17

**Authors:** Ciro Gioia

**Affiliations:** Independent Researcher, Via Roma 34b, 21020 Brebbia, Italy; cirogioia@tin.it

**Keywords:** timing, T-RAIM, Galileo, timing retrieval, integrity

## Abstract

Several applications rely on time retrieved from Global Navigation Satellite System (GNSS), and this pushes for integrity tailored to timing. Integrity information could be broadcast by GNSS itself, but currently, there are no GNSSs providing such integrity information for a timing application. The integrity provided by GNSS itself could not be timely enough for real time users and does not include local effects due to multipath or other local interferences. In order to fill the gap, integrity can be locally/autonomously computed by the receiver using Timing Receiver Autonomous Integrity Monitoring (T-RAIM) algorithms. Three T-RAIM algorithms have been designed, implemented, and tested; specifically, the algorithms are Forward-Backward (FB), Danish, and Subset. The algorithms are applied to the classical Position Velocity and Timing (PVT) solution and to the time-only case assuming the receiver coordinates are known. Tests using two identical receivers located in different scenarios, open-sky and obstructed, have been carried out to validate the algorithms proposed. The increased redundancy obtained from the knowledge of the receiver coordinates play a fundamental role for the integrity algorithms performance. The benefits of the T-RAIM algorithms activation, in signal degraded conditions, clearly emerged in terms of frequency error and Allan Deviation (ADEV). A small increase of the execution time has been observed when the T-RAIM algorithms are used.

## 1. Introduction

In recent years, the exploitation of GNSS for timing purposes has increased significantly. Currently, several applications are relying on GNSS-based timing, including critical infrastructures in different sectors such as telecom, energy, and finance. The requirements of these applications are usually defined in terms of maximum error on the Pulse Per Second (PPS) (either with respect to GNSS Time (GNSST) or Universal Time Coordinated (UTC)) and depending on the application [1]. The typical accuracy required in telecom, energy, finance, is of the order of 1 microsecond ($10^{-6}$ s); but the market evolution and the large application of GNSS-based timing and synchronization function in the 5G network operations will make this value more stringent [2]. As reported in [3], applications such as smart grids (for electricity transmission) or 5G (in telecom) might require an accuracy of 100 ns or better. Finally, some specific scientific applications (e.g., astronomical interferometry) could require an accuracy of a few nanoseconds ($10^{-9}$ s), which is at the edge of the accuracy obtainable from GNSS.

A large part of the applications relying on GNSS time and synchronization are performed in open-sky conditions where the accuracy of the timing solution is mainly influenced by the satellite availability and by the geometry that is represented by the Time Transfer Dilution Of Precision (TTDOP) [4]. In the open-sky scenarios, the accuracy of the timing solution could be estimated as the product of the User Equivalent Ranging Error (UERE) and the TTDOP; for a single frequency user the UERE is in the order of 6 m [4]. With these values, an accuracy in the nanoseconds order is obtained and it is fully aligned with the requirements mentioned above.

In the periodic forum organized by European Union Agency for the Space Programme (EUSPA), users from different market segments discuss their needs and application-level requirements relevant for Position, Navigation and Timing (PNT) [5]. During the section related to the critical infrastructure, the need for integrity information tailored to the timing solution was discussed [6]. The integrity concept discussed was broader than that used for safety-critical or civil aviation encompassing concepts of quality assurance/quality control.

Integrity information could be generated and broadcast by GNSS itself; new services related to timing integrity are under discussion, for example, dedicated Timing Service Message (TSM), including indicators (e.g., flags), to increase the trust on the timing solutions will be broadcast by Galileo in its Open Service (OS) navigation message [7]. Currently, there are no GNSSs providing such integrity information for a timing application. In addition, the integrity information broadcast by the GNSS itself is not timely enough for real time users and does not include local effects due to multipath or other local interference [8]. In order to fill these gaps, integrity information can be locally/autonomously computed by the receiver using T-RAIM algorithms. The T-RAIM has been defined as a fundamental element of the future European GNSS timing service [9], and the integrity algorithms are also considered in the new standard for Galileo timing receivers proposed by the European Commission (EC) [10]. In particular, the following T-RAIM processing characteristics were requested:Using a performance-based approach to give the manufacturer’s the freedom of implementation;Static receiver and dynamic receiver options: considering the cases with and without known receiver position.

Very little information is available on the T-RAIM algorithms, Ref. [11] addresses the time integrity issue of GPS, introduced an algorithm tested under unusual satellite conditions, but few details are missing for the proper implementation of the algorithm. Motorola developed a T-RAIM algorithm (ONCORE) [12], able to provide integrity information for GPS timing but the details of the algorithm are not publicly available. Hence, one of the goals of this paper is to illustrate the possible strategies for T-RAIM algorithms.

In this paper, three different T-RAIM algorithms have been designed, implemented, and tested; specifically, the algorithms are FB [13], Danish, and Subset. A detailed description of the implementation is provided in Section 2. The approaches are derived from classical RAIM algorithms, described in Refs. [14,15].

In order to verify the impact of the implemented algorithm, a specific set-up has been designed. The algorithms have been tested using live real signals collected by two identical receivers placed at a distance of about 50 m. One of the receiver was placed in open-sky conditions while the other was placed in obstructed conditions. A more detailed description of the set-up and of the data collected is provided in Section 4.

The analysis focused on Galileo single frequency case. From the results, a light increase of the computation time emerged when the T-RAIM algorithms were activated in both scenarios. In the obstructed scenario, all the proposed T-RAIM algorithms provided a similar reliable availability. In addition, a clear reduction of the frequency error has been observed when T-RAIM is activated.

The remainder of the paper is structured as follows: in Section 2, the computation of the timing solution is presented together with the description of the integrity algorithms. In Section 3, the metrics used to assess the performance of the algorithms are detailed; the experimental set-up and the data collected are described in Section 4. The experimental results are discussed in Section 5. Finally, Section 6 concludes the paper.

## 2. Timing Solution

This section describes the timing solution and the integrity block including the three T-RAIM algorithms.

### 2.1. Timing Solution Estimation

In this work, the timing solution is computed using code measurements, i.e., pseudorange, on the Galileo E1 signal. Two different strategies are used to compute the clock parameters depending on the knowledge of the receiver coordinates. If the receiver coordinates are unknown, a traditional Single Point Positioning (SPP) approach is used [8,16,17]; while, if the receiver coordinates are known, a timing solution is computed as follows.

The raw pseudoranges derived from RINEX files are corrected as: (1)$$PR_{corr}^{i}=PR_{raw}^{i}+clock_{sat}^{i}+rel_{eff}^{i}-TGD^{i}-Iono^{i}-Tropo^{i}-d^{i}$$
where $PR_{raw}^{i}$ are the raw measurements reported by the receiver for the ith satellite, $clock_{sat}^{i}$ is the satellite clock error corrected using the parameters broadcast in the Galileo navigation message [4], $rel_{eff}^{i}$ is the relativistic effect including Sagnac effects [8], $TGD^{i}$ is the Time Group Delay (TGD) corrected using the parameter in the navigation message [18], $Iono^{i}$ is the ionospheric delay corrected using the Klobuchar model [19], $Tropo^{i}$ is the troposphere error mitigated using Saastamoinen model [20]. $d^{i}$ is the distance between the receiver and satellite and it is removed using the a priori information on the receiver coordinates.

Finally, corrected pseudoranges are used to compute the timing parameters using a Weighted Least Squares (WLS) approach
(2)$$rx_{clockBias}={\left(H^{T}\cdot W\right)}^{-1}\cdot H^{T}\cdot W\cdot {\mathrm{PR}}_{\mathrm{corr}}$$
$rx_{clockBias}$ is the receiver clock bias, H is the design matrix composed of a single column of one, W is the weighting matrix, with weights related to satellite elevation [14,21], ${\mathrm{PR}}_{\mathrm{corr}}$ is the vector containing the corrected pseudoranges from all the satellites.

The residuals *r* are defined as: (3)$$r={\mathrm{PR}}_{\mathrm{corr}}-H\cdot rx_{clockBias}.$$

### 2.2. Integrity Algorithms

Some of the applications relying on GNSS-based timing are performed in environments where the signals are affected by errors due to multipath, fading, etc.; these phenomena could lead to large errors in the final timing solutions. In order to limit the impact of gross errors in the timing solutions, a specific block providing integrity information on the timing solution is required and is herein called the integrity block [9]. The block does not only verify if the solution is trustable but it also verifies the consistency of the whole measurement set, and eventually searches, identifies, and excludes the outliers [22]. The integrity block is integrated in the navigation solution and all the checks are performed after the estimation process, as shown in Figure 1; the light blue box includes all the additional steps needed for the integrity algorithms. In this work, three algorithms are considered for the integrity block, and the algorithms are described in the following sections.

#### 2.2.1. Forward-Backward

The FB scheme foresees two main phases: the forward and backward [15,22,23,24]. In the first phase, the geometry of the system is verified. The geometry check is based on the Time Protection Level (TPL), computed as: (4)$$TPL=WATP+TPL_{noise}$$
where WATP is the Weighted Approximated Time Protected (WATP), which contains the information on the measurements biases; while the $TPL_{noise}$ is the term representing the measurements noise [25].

The WATP is based on the Timing Slope (TS), in particular on its weighted version Weighted Timing Slope (WTS), which computes as: (5)$$WTS_{i}=\frac{a_{i}}{S_{i}}$$
where $a_{i}$ is the ith element of the vector a (see Equation (Equation 5)) and *S* (see Equation (Equation 6)) is the slope matrix.
(6)$$a={\left(H^{T}\cdot W\cdot H\right)}^{-1}\cdot H\cdot W$$
(7)S=I−H·a
where I is the identity matrix.

The maximum value of the WTS is used to compute the WATP: (8)$$WATP=maximum\left(\mathrm{WTS}\right)\cdot p_{bias}$$
$p_{bias}$ is the square root of the Global Test (GT) threshold [26].

The term related to the measurements noise is computed as: (9)$$TPL_{noise}=K\cdot \sqrt{\sigma_{sol}^{2}}$$
*K* is the protection level coefficient related to the probability of missed detection [27], and $\sigma_{sol}^{2}$ is the variance of the timing solution.

If the TPL is lower than the Time Alarm Limit (TAL), the geometric conditions are met and the whole measurement set is analyzed using a GT, and the test is based on the residuals defined in Equation (Equation 3). If an inconsistency among the measurements is detected, a Local Test (LT) is carried out to identify the possible outlier. The measurement identified as the possible outlier is excluded after a separability check based on the correlation coefficient among the measurements. The first phase is repeated until no more outliers are detected. If more than one measurement has been excluded in the forward phase, the backward phase is performed. This second block has the main function of re-introducing the measurement wrongly excluded and it is based only on the GT. The processing scheme of the FB algorithm is shown in Figure 2. A more complete description of the algorithm is available in Refs. [13,14]. In this algorithm, the solution can be declared unreliable for three reasons:Geometry; the system is not robust enough to support integrity checks;Inconsistency between GT and LT; GT detects an inconsistency among the measurements set but all the measurements pass the LT;Separability; LT identifies a measurement as an outlier but it is too correlated with other measurements and it is not possible to identify the real outlier.

#### 2.2.2. Danish

The Danish algorithm is very similar to the forward phase of the FB algorithm and it exploits the very same elements of the FB algorithm: TAL check, GT, LT, and Separability [28,29]. Between the two algorithms, two main differences can be noted: the first one is the absence of the backward phase in the Danish algorithm; the second difference is how the outlier is treated. In the Danish case, the measurement is not excluded but it is iteratively de-weighted; an extreme de-weighting of the measurement is equivalent to the measurement exclusion. In addition, for the Danish algorithm, the solution is declared unreliable for the same three cases of the FB algorithm. The processing scheme of the Danish algorithm is described in Figure 3.

#### 2.2.3. Subset

The elements of the Subset algorithm are: the geometry check and the GT [15,30]. If the geometry of the system is robust enough and an inconsistency among the measurements is detected, then all the possible subsets obtained excluding one measurement are checked. If one or more subsets pass the GT, the solution is declared reliable and it is computed with the subset passing the GT and with the lowest decision threshold value. The process is performed iteratively excluding up to half of the available measurements. Using this scheme, the solution can be declared unreliable only for geometrical reason. The scheme of the algorithm is shown in Figure 4.

## 3. Performance Metrics

In this section, the parameters used to assess the performance of the algorithms are presented.

The performance is assessed in terms of:Reliable availability, defined as: the percentage of time in which the timing solution is declared reliable by the integrity algorithm. The reliable availability is computed as:
(10)$$RelAvail=100\cdot \frac{RelEpoc}{TotNumEpoc}$$
where RelEpoc is the number of the reliable epochs, and TotNumEpoc is the total number of epochs of the dataset.Frequency error: represents the variations of a timing signal generated by a clock. Here the frequency error is estimated from the receiver clock bias estimation according to the method described in Ref. [21]. The frequency error is computed as:
(11)$$Fre_{err}\left[t\right]=\frac{rx_{clockBias}\left[t\right]-rx_{clockBias}\left[t-K\right]}{\tau }$$
where $Fre_{err}\left[t\right]$ is the normalized frequency error at the epoch *t*, $rx_{clockBias}\left[t\right]$ is the receiver clock bias estimation in seconds, *K* is an integer, and τ is the averaging time interval.
(12)$$\tau =K\cdot dt_{data}$$
in which $dt_{data}$ is the sampling rate of the receiver clock bias.Allan Deviation: the square root of the Allan variance, which is a generalization of the sample variance and is commonly used to characterize the stability of oscillators [31]. This parameter provides indications about the expected frequency deviation that can occur in the averaging time interval, τ [32]. Additional details for the ADEV estimation using GNSS measurements are available in Ref. [21].Execution time: the time needed to execute the algorithm. In particular, the total execution time (the time needed for processing the whole dataset) and the single epoch execution time are evaluated. This parameter provides an idea of the computational load required by the receiver to implement a specific algorithm.

## 4. Experimental Setup

The algorithms developed have been tested using real live signals. For the setup, two identical Septentrio Mosaic 5 [33] receivers were used. The receivers were placed in two different locations at a distance of about 50 m. The first receiver was placed in open-sky conditions, connected to an antenna placed on the roof of the European Microwave Signature Laboratory (EMSL) at the Ispra site of the Joint Research Centre (JRC). The second receiver was placed in an obstructed scenario just outside the EMSL building; the receiver was surrounded by trees (attenuating GNSS signals) and tall buildings (introducing multipath). The two devices simultaneously logged data for about 4 h. The two different scenarios allowed the assessment of the algorithms in different conditions; in particular, the second scenario is used to asses the capability of the T-RAIM algorithms to deal with multiple outliers. The coordinates of the antennas are reported in Table 1.

The coordinates of the antenna in open-sky conditions were computed using the Precise Point Positioning (PPP) algorithm with a very long data series; a more detailed description of the antenna location is available in Ref. [34]. In the obstructed scenario, the signal degradation was so severe that it did not allow the use of the PPP approach; hence, the coordinates of the antenna were computed using a total station. The accuracy of the timing solution is strictly related to the satellite availability (i.e., the number of visible satellites) and to the TTDOP. In order to represent the difference between the two considered scenarios, the statistical parameters (minimum, maximum, and average) of the number of tracked satellites and TTDOP are reported in Table 2. From the table, it emerges that the number of tracked satellites is quite different for the two scenarios: the mean number of satellites used in the timing solution is reduced to about three, passing from open-sky to obstructed scenario. The minimum number of available satellites is 2 and 5 for the obstructed and open-sky scenarios, respectively. The reduced number of satellites impact directly the TTDOP, this is reflected in the parameters reported in Table 2, where all the parameters for the TTDOP are higher for the obstructed case.

The number of visible satellites for the two datasets is shown in Figure 5. From the figure, it clearly emerges that in the obstructed scenario, the number of visible satellites is limited. In particular, after about one hour from the start of the test, only four satellites were available, hence the reliable availability of the PVT solution is limited. In the same time frame in the open sky, the number of satellites varied between 7 and 5.

In addition to the number of satellites, the quality of the measurements is a fundamental element for the accuracy of the timing solution. To represent the different scenario conditions, the distribution of the multipath errors in the two scenarios is shown in Figure 6. From the figure, it can be noted the multipath errors are larger in the obstructed scenarios. So, the obstructed scenario is characterized by a reduced number of satellites and large errors in the measurements, making it a very challenging scenario for T-RAIM.

## 5. Results

In this section, the results applying the different T-RAIM algorithms are presented.

The execution time for the different algorithm is shown in Figure 7. From the figure, it can be noted that the activation of the T-RAIM leads to an increase of the execution time; without the integrity checks the baseline algorithm needs about 87 and 82 s to process the open-sky and obstructed datasets, respectively. When T-RAIM is activated, the total execution time increases up to 135 s for the Danish algorithm in obstructed conditions. Comparing the results in the two scenarios, it can be noted that in the obstructed scenario the execution time is always larger than in open-sky conditions. This is due to the higher number of outliers to be detected and mitigated. Similar results are obtained considering the single epoch execution time.

The reliable availability obtained using the three T-RAIM algorithms in the two scenarios considering the full PVT solution and the time-only solution are shown in Figure 8. From the figure, it can be noted that the lowest reliable availability is obtained in the obstructed scenario when the coordinates of the receivers are not known and a minimum of five satellites is required to perform the integrity checks. This low value is due to the limited number of satellites available in the scenario as shown in the second row of Table 2; in the same conditions, when the receiver position is known and only the timing parameters need to be estimated (only two satellites are needed for integrity checks), the reliable availability is higher than 90% in all cases.

The TPLs for the two cases, obstructed and open sky, are shown in Figure 9. The blue line represents the open sky conditions; while, the red line is linked to the obstructed scenario. Finally, the TAL is the black dashed lines. From the figure, it can be noted that the red line is always higher than the blue one. This is due to the lower number of visible satellites in the obstructed scenario. Both lines are well below the TAL set to 30 ns.

In order to analyze the impact of the exclusions performed by the T-RAIM algorithms on the clock bias estimation, the difference between the clock bias estimated with and without T-RAIM is computed: (13)$$\Delta_{ClockBias}\left[t\right]=rx_{clockBias}^{NoT-RAIM}\left[t\right]-rx_{clockBias}^{T-RAIM}\left[t\right]$$
where $rx_{clockBias}^{NoT-RAIM}$ is the receiver clock bias at epoch *t* without T-RAIM and $rx_{clockBias}^{T-RAIM}$ is the receiver clock bias estimated with one of the three T-RAIM algorithms activated.

The clock bias difference as a function of the time is shown in Figure 10, considering open-sky (red lines) and obstructed (blue lines) scenarios. The full PVT strategy is sued for the clock bias estimation; in each box a T-RAIM algorithm is considered. From the figure, it clearly emerges that the limited number of available satellites limits the application of the T-RAIM algorithms and the effects are visible only in the first part of the dataset when more satellites were available. For the FB case, the maximum difference in the estimated clock bias is about 10 m (about 30 ns); for the Danish and subset case the maximum difference is about 20 m (about 60 ns).

In Figure 11, the clock bias difference as a function of time considering the time-only strategy is shown. The three T-RAIM algorithms are considered separately in the three boxes. In open-sky conditions (red line), no impact can be noted because no exclusions are performed by the three algorithms. Meanwhile, in the obstructed scenario the impact of the T-RAIM exclusions is evident in the whole dataset. The impact of the three algorithms is very similar and only small differences can be noted. This shows that the algorithms identify and excluded/de-weighted the same satellites. The maximum value of the $\Delta_{ClockBias}$ is about 25 m (80 ns). This is observed toward the end of the data collection when less satellites were available and the outliers’ effects were more evident for the no T-RAIM case, as the value is common to all the three algorithms.

In order to evaluate the impact of the T-RAIM algorithms on the frequency error, frequency error evolution with and without T-RAIM considering full PVT and time-only strategies is shown in Figure 12. Only results for the obstructed scenario are shown because no differences can be appreciated in open-sky. From the figure, it can be noted that the configuration with the largest frequency error variations is the one using the full PVT strategy without T-RAIM (blu line). For this strategy, the activation of the T-RAIM leads to a reduction of some of the peaks in the frequency error in the first part of the dataset; comparing the blue (PVT No T-RAIM) and red-dashed (PVT T-RAIM) lines, the Danish and Subset schemes seem to be more effective for the FB in these conditions. A large reduction of the frequency error can be noted to be passing from the full PVT strategy (blue line) to the time-only solution (yellow line) but still some spikes are visible in the frequency error. The application of the T-RAIM algorithms to the time-only cases further reduced the spikes: the purple dashed line is the one with less spikes.

In order to evaluate the impact of the T-RAIM algorithms on the clock stability, the ADEV of the clock bias estimate with and without T-RAIM is shown in Figure 13; as the sample case, the FB algorithm is considered. In the left box, all the solutions are considered: hence, for the configurations with T-RAIM activated, the solutions flagged as unreliable are also used for the estimation of the ADEV. In the right box, only reliable solutions are considered (also for the configurations without T-RAIM). When all solutions are considered, the benefits of the T-RAIM algorithms can be appreciated only on the full PVT solution. While for the time-only case, only small differences can be noted. The curves obtained considering only reliable solutions are lower than in the case with all the solutions.

In Figure 14, the ADEVs computed using the three different T-RAIM schemes are shown for both solution types full PVT (left box) and time-only (right box); the ADEVs are computed considering only reliable epochs in the obstructed scenario. Among the different schemes, only marginal differences can be appreciated; in particular, for an averaging time interval smaller than 20 s, the Danish method provides the higher stability for both types of solutions. For averaging the time interval between 20 and 200 s, the FB scheme is the one with the lowest values for the time-only solution case; in the same averaging time interval for the full PVT solution, the subset scheme provides the highest clock stability. Finally, for an averaging time interval larger than 200 s, all the schemes have identical performance.

## 6. Conclusions

This paper presents three T-RAIM algorithms, namely FB, Danish, and Subset. The algorithms have been designed, implemented, and tested using a real live signal from Galileo. Two different strategies for the computation of the clock parameters are also proposed, depending on the receiver coordinates knowledge.

The algorithms have been tested with a dedicated set-up: two identical Mosaic receivers were used, the receivers were placed in two different scenarios: one in open-sky and the other in obstructed conditions.

The algorithms have been assessed in terms of execution time, reliable availability, frequency error, and ADEV.

From the results, it emerged that the inclusion of a T-RAIM algorithm increases the complexity of the navigation algorithm. leading to an increased execution time in both scenarios, the increase is more evident in the obstructed scenarios where the presence of multiple outliers leads to multiple iterations of the T-RAIM algorithms.

T-RAIM algorithms are strongly dependent on the redundancy, hence the knowledge of the receiver coordinates significantly improve the performance of the algorithms. In particular, in obstructed conditions, a strong reduction of the reliable availability (down to 20%) has been observed when the full PVT solution is used for estimating the clock parameters; while, when the time-only solution is used, the reliable availability is higher than 90%. This is due to the increased redundancy, which increases the capacity of the T-RAIM algorithms to identify and reject multiple outliers.

In the frequency error domain, it emerged that the configuration with the largest frequency error variations is the one using the full PVT strategy without T-RAIM. When T-RAIM is activated, a reduction of some of the peaks in the frequency error has been observed. Danish and Subset schemes seems to be more effective of the FB in the considered scenarios. A large reduction of the frequency error has been noted passing from the full PVT strategy to the time-only solution. The application of the T-RAIM algorithms to the time-only cases further reduced the spikes.

From the analysis of the clock stability, it can been noted that among the different T-RAIM schemes only marginal differences can be appreciated:For averaging time interval smaller than 20 s, the Danish method provides the highest stability;For averaging time interval between 20 and 200 s the FB scheme is the one with the lowest ADEV values for the time-only solution case;For averaging time interval between 20 and 200 s for the full PVT solution the subset scheme provides the highest clock stability.

## Figures and Tables

**Figure 1 sensors-23-02283-f001:**
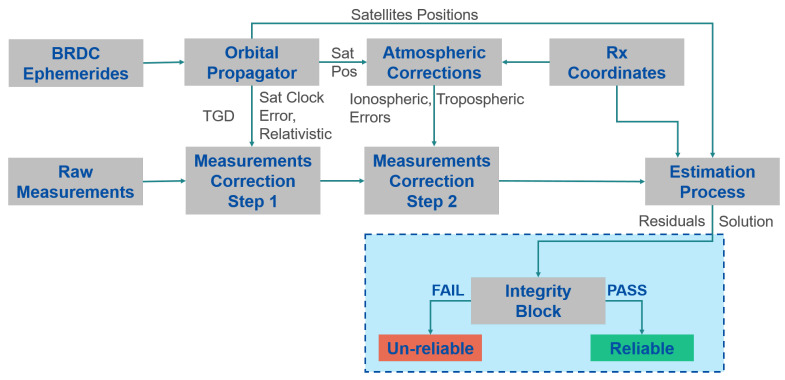
Diagram of the algorithm used for the computation of the timing solution.

**Figure 2 sensors-23-02283-f002:**
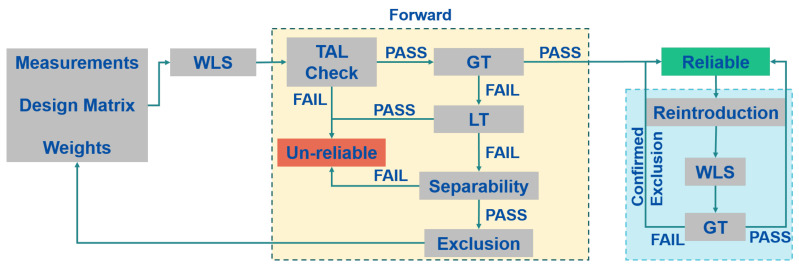
Scheme of the T-RAIM Forward-Backward algorithm.

**Figure 3 sensors-23-02283-f003:**
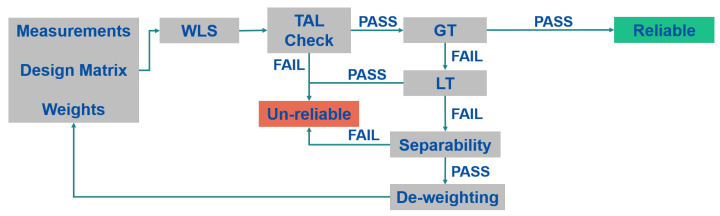
Scheme of the T-RAIM Danish algorithm.

**Figure 4 sensors-23-02283-f004:**
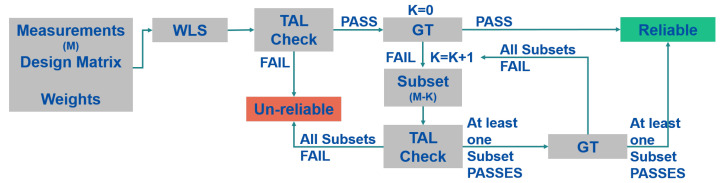
Scheme of the T-RAIM Subset algorithm.

**Figure 5 sensors-23-02283-f005:**
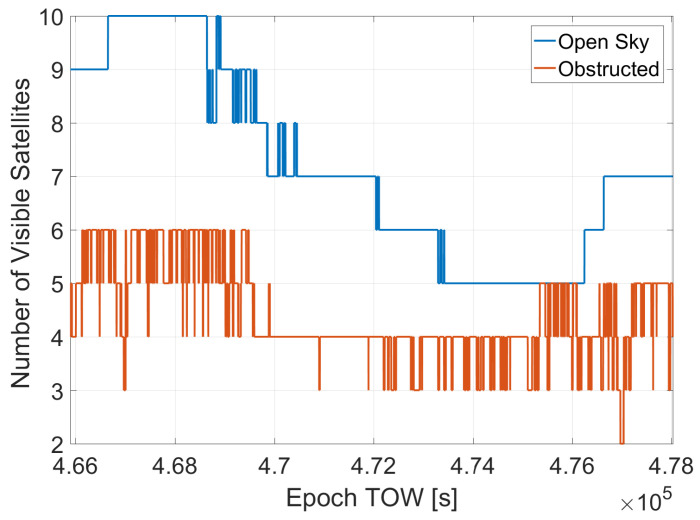
Number of visible satellites for the two scenarios.

**Figure 6 sensors-23-02283-f006:**
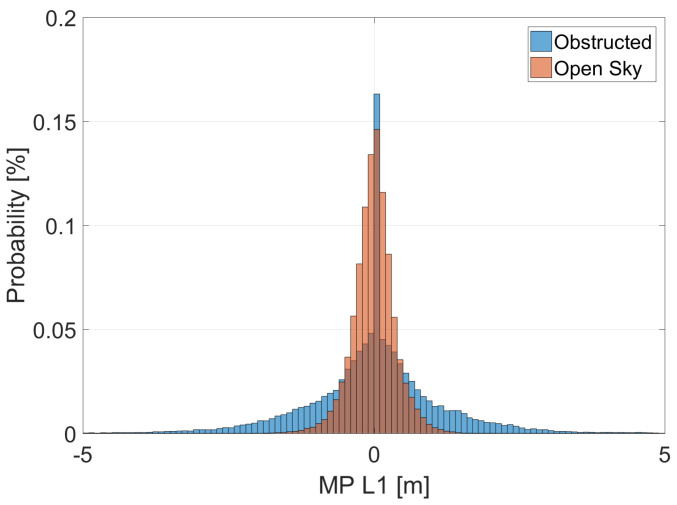
Distribution of the Multipath errors for the two scenarios.

**Figure 7 sensors-23-02283-f007:**
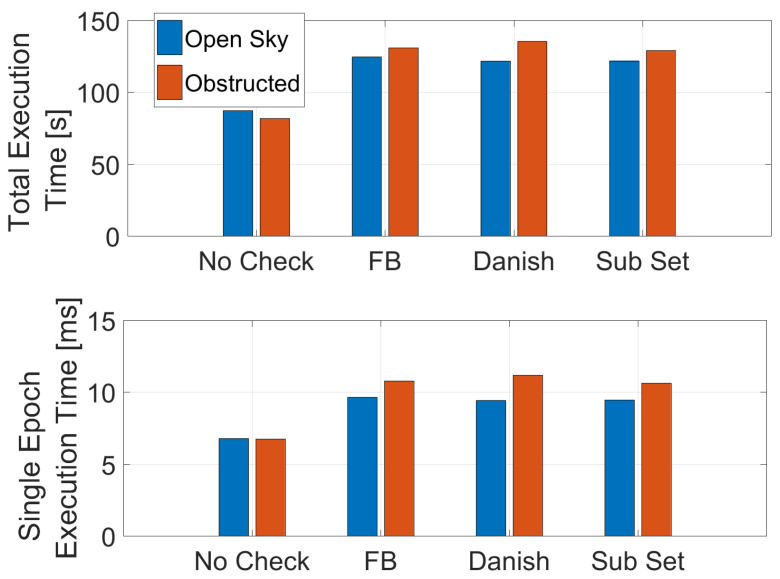
Execution time for the different algorithms in the two scenarios considered.

**Figure 8 sensors-23-02283-f008:**
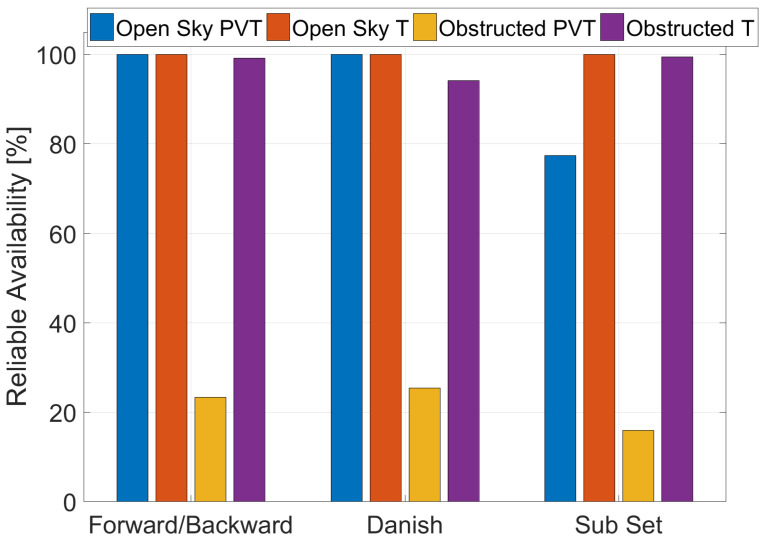
Reliable availability values for the three T-RAIM algorithms in the two scenarios considering full PVT and time-only solutions.

**Figure 9 sensors-23-02283-f009:**
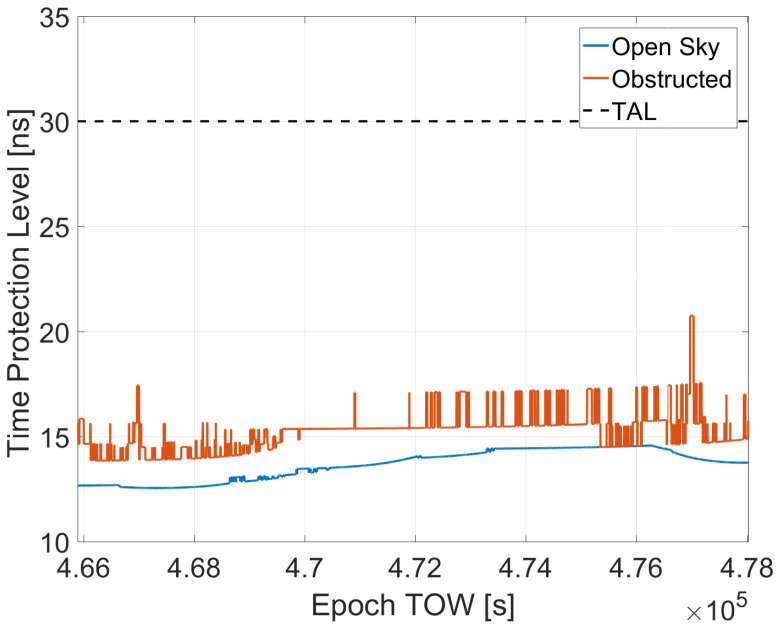
TPLs for the two cases: blue line open sky; red line obstructed.

**Figure 10 sensors-23-02283-f010:**
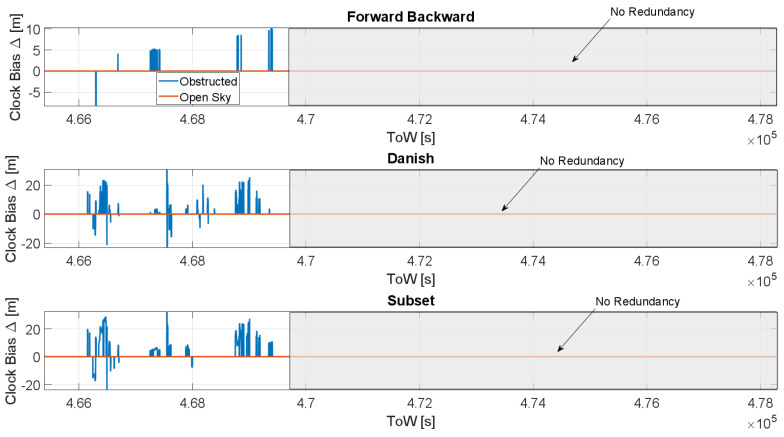
Difference between the clock bias estimation with and without T-RAIM considering the full PVT strategy.

**Figure 11 sensors-23-02283-f011:**
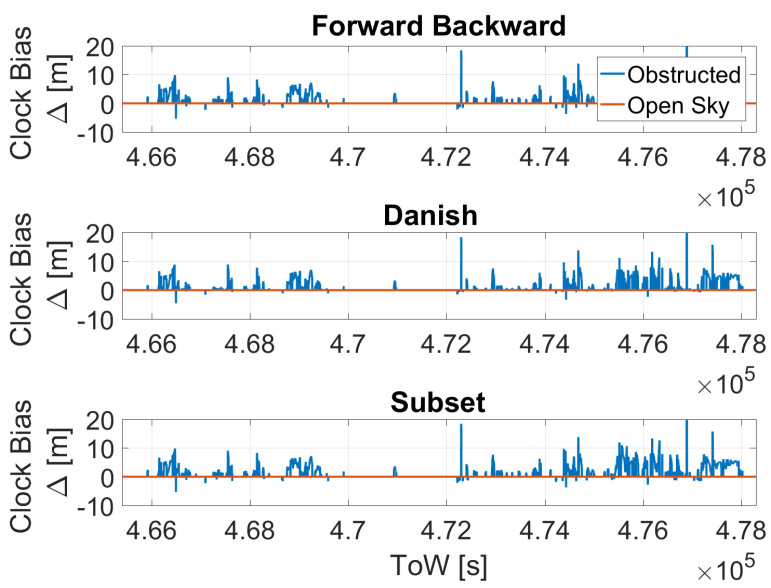
Difference between the clock bias estimation with and without T-RAIM considering the time-only strategy.

**Figure 12 sensors-23-02283-f012:**
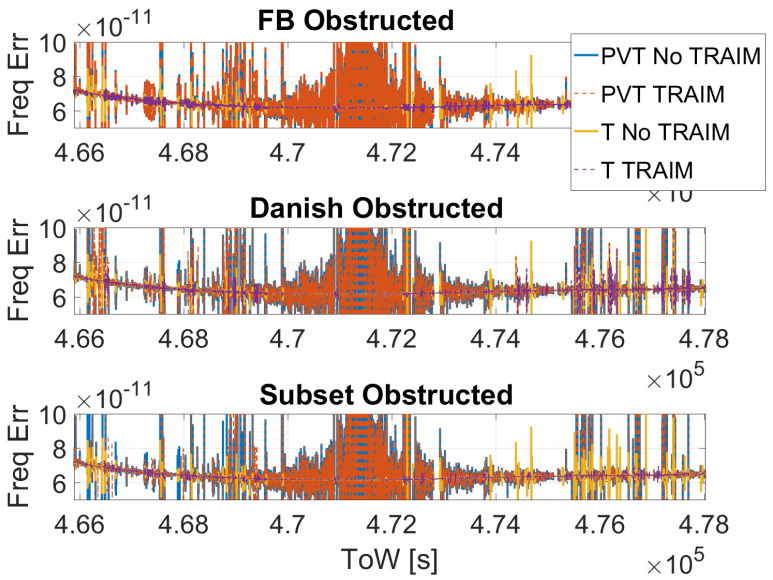
Frequency error as a function of time considering the configurations with and without T-RAIM in the obstructed scenario.

**Figure 13 sensors-23-02283-f013:**
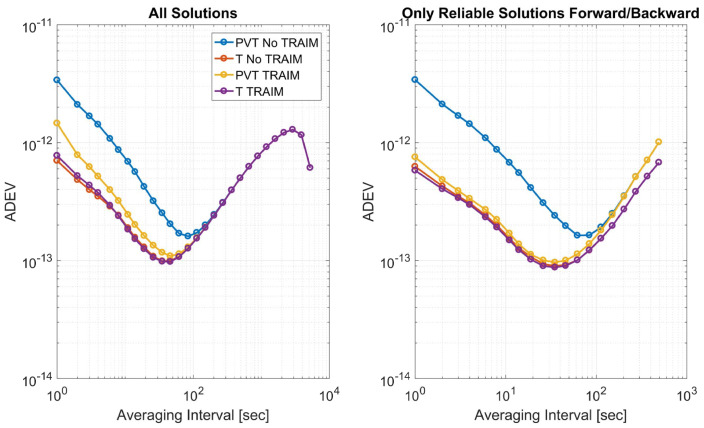
Allan Deviations of the configurations with and without T-RAIM considering the FB scheme.

**Figure 14 sensors-23-02283-f014:**
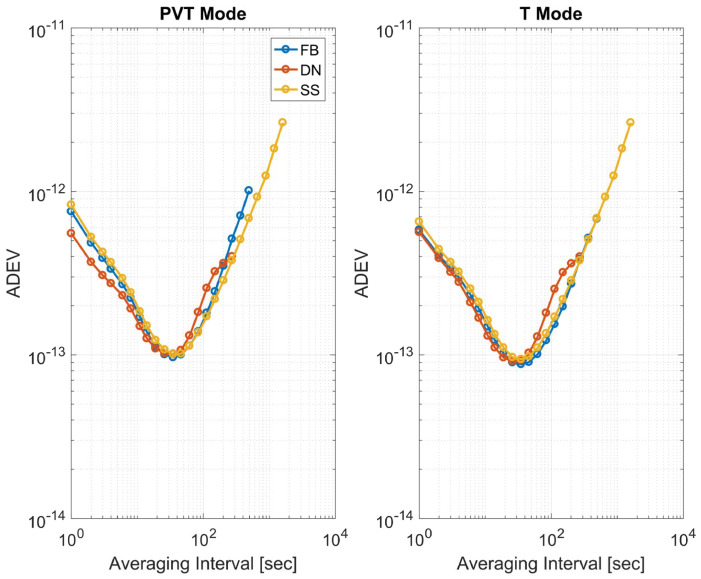
ADEV of the clock bias estimate for the three T-RAIM schemes considering only reliable solutions for the obstructed scenario.

**Table 1 sensors-23-02283-t001:** Coordinates of the antennas used for the test.

Scenario	Latitude [deg]	Longitude [deg]	Height [m]
Open-Sky	45.8104	8.6300	279.1650
Obstructed	45.8104	8.6302	259.1840

**Table 2 sensors-23-02283-t002:** Available satellites and geometric conditions for the two scenarios.

Scenario	Num Satellites			TTDOP		
	Mean	Min	Max	Mean	Min	Max
Open-Sky	7.24	5	10	0.38	0.32	0.45
Obstructed	4.45	2	6	0.48	0.41	0.71

## Data Availability

Not applicable.

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
