# Peer review of "T-RAIM Approaches: Testing with Galileo Measurements"

_sensors, 2023, doi:10.3390/s23042283_

Round 1
Reviewer 1 Report
The paper deals with the analysis of three Timing Receiver Autonomous Integrity (T-RAIM) algorithms, designed for Global Navigation Satellite System (GNSS) receivers. The algorithms are tested in open sky and obstructed scenarios, and applied to both classical Position Velocity and Timing (PVT) solution and time only case. The manuscript is clearly written and well organized. It describes the integrity algorithms, along with the experimental set-ups and analyses the achievable performance.
As a general comment, it might be good to include a short paragraph mentioning how the three algorithms presented in the paper can be compared and/or referred to state-of-the art solutions.
Hereafter some further minor comments that might be useful to improve the quality of the paper:
- Figures legends and axes labels are too small and very difficult to read
- Figures 8 and 9 can be better understand by adding a figure with the number of visible satellites over time
- Figure 8: the same y-axis scale shall be used for the three subplots
- Lines 64-66: do not detail the contributions of each sub-sub-section
- Line 119-120: the sentence needs to be rephrased
Author Response
The author wish to thank the reviewer for the thorough and useful evaluation of the paper. The author has benefited from the reviewer comments and insights and have revised the paper according to the suggestions.
Detailed replies to the reviewers’ comments are provided in the attached file.

Reviewer 2 Report
This manuscript investigated three Timing Receiver Autonomous Integrity Monitoring (T-RAIM) algorithms, i.e., Forward-Backward (FB), Danish and Subset. The three T-RAIM algorithms have been designed, implemented and tested with Galileo measurements.
1) This manuscript used four metrics (reliable availability, frequency error, Allan Deviation, and execution time) to assess the performance of the three algorithms. However, for integrity monitoring, parameters such as alert limit, time to alert, integrity risk, protection level, should be assessed for the proposed algorithms.
2) There is only one subsection (4.1) in Section 4, so the title of Subsection 4.1 could be deleted.
3) In Section 5 Results, a table could be used to compare the performance of the three algorithms.
4) The conclusion from this research should be given in Abstract.
5) The equations should be given with formal format, e.g., vector and matrix should be in Bold.
6) References for three algorithms should be cited in the manuscript.
Author Response

(The authors gave the same response as above.)
